# Decentralized Multi-Armed Bandit Can Outperform Classic Upper Confidence Bound

## Abstract

This paper studies a decentralized multi-armed bandit problem in a multi-agent network. The problem is simultaneously solved by $N$ agents assuming they face a common set of $M$ arms and share the same mean of each arm's reward. Each agent can receive information only from its neighbors, where the neighbor relations among the agents are described by a directed graph whose vertices represent agents and whose directed edges depict neighbor relations. A fully decentralized multi-armed bandit algorithm is proposed for each agent, which twists the classic consensus algorithm and upper confidence bound (UCB) algorithm. It is shown that the algorithm guarantees each agent to achieve a better logarithmic asymptotic regret than the classic UCB provided the neighbor graph is strongly connected. The regret can be further improved if the neighbor graph is undirected.

## 1 Introduction

Multi-armed bandit (MAB) is a basic but fundamental reinforcement learning problem which has a wide range of applications in natural and engineered systems including clinical trials, adaptive routing, cognitive radio networks, and online recommendation systems [1]. The problem has various formulations. In a classical and conventional MAB problem setting, a single decision maker (or player) makes a sequential decision to select one arm at each discrete time from a given finite set of arms (or choices) and then receives a reward corresponding to the chosen arm, generated according to a random variable with an unknown distribution. In general, different arms have different distributions and reward means. The target of the decision maker is to minimize its cumulative expected regret, i.e., the difference between the decision maker's accumulated (expected) reward and the maximum which could have been obtained had the reward information been known. For this conventional MAB problem, both lower and upper bounds on the asymptotic regret were derived in the seminal work [2], and classic UCB algorithms were proposed in [3] which achieve an $O(\log T)$ regret. Since multi-armed bandits have been studied for decades, it is impossible to survey the entire bandit literature here. For an introductory survey, see a recent book [4].

Over the past decade, our social networks, communication infrastructure, data centers, and societal systems have become increasingly massive and complex, which can all be modeled as networked multi-agent systems. In such a large-scale multi-agent network, e.g. a sensor network and a multi-robot system, the need for decentralized information processing and decision making arises naturally since the sensors or robots in the network are equipped with on-board processors and are physically separated from each other. Concurrently, the emerging big data era brings restrictions on information flow to human-involved networks, primarily due to privacy concerns, and thus precludes conventional centralized and parallel information processing and decision making algorithms, which typically rely on a center collecting all information or taking the lead. Therefore, there is ample motivation to develop multi-agent, decentralized, multi-armed bandit algorithms.

Over the past year, there has been increasing interest to extend conventional single-player bandit settings to multi-player frameworks. Notable examples include [5–17], to name a few. Among all the existing multi-agent settings, we are motivated by a cooperative setting which makes use of a consensus process [18, 19] among all agents. Such a setting was first proposed in [16] with homogeneous reward distributions, i.e., all agents share the same distribution of each arm's reward. The problem has recently attracted increasing attention and quite a few different consensus-based

decentralized algorithms have been proposed and developed [16, 17, 20–23]. Note that in such a homogeneous reward distribution setting, each agent in a network actually can independently learn an optimal arm using any conventional single-agent UCB algorithm, ignoring any information received from other agents. Notwithstanding, all the existing algorithms for the decentralized multi-armed bandit problem with homogeneous reward distributions require that each agent be aware of certain network-wise global information, such as spectral properties of the underlying graph or total number of agents in the network. Such a requirement leads to a counterintuitive observation: compared with the conventional single agent case, each agent in a multi-agent network can collect more arm-related information while its bandit learning becomes more restrictive or less independent. Motivated by this issue, this paper aims to develop a fully decentralized multi-armed bandit algorithm for a general directed graph, which does not require any global information, and further shows that the decentralized algorithm can ensure each agent in the network learns faster in contrast to the conventional single-agent case.

**Related Work**   Multi-agent MAB problems have been studied in various settings [5–17]. For example, [5, 6, 9, 24] preclude communications among agents but allow them to receive "collision" signals when more than one agent selects the same arm, which has applications in wireless communication and cognitive radio. A distributed setting with a central controller is studied in [13, 25] in a federated learning context. Other federated bandit settings are considered in [12, 23, 26] with additional focus on theoretical privacy preservation.

Consensus-based decentralized MAB algorithms are developed in [16, 17, 20–23] for homogeneous reward distributions in a cooperative multi-agent setting. Very recently, cooperative multi-agent bandits have been extended to heterogeneous reward settings, that is, different agents may have different reward distributions and means for each arm. A heterogeneous decentralized problem is solved in [23] using the idea of gossiping to improve communication efficiency and privacy. All these consensus- or gossip-based MAB algorithms require global information. An exception is [27] which considers a heterogeneous setting but focuses on a complete graph, which implicitly allows each agent to collect all other agents' information.

**Technical Challenges**   The design of a suitable upper confidence bound function is a critical step in crafting a multi-armed bandit algorithm for the conventional single-agent case, which determines the decision of which arm to choose at each time and thus plays an important role in quantifying the cumulative regret. Such a relationship between upper confidence bound and regret becomes much more complicated in the decentralized setting because with the agents' information propagating over the network, each individual agent's regret is coupled with all the other agents' upper confidence bound functions. This is likely the reason why all the existing algorithms for a similar decentralized multi-armed bandit problem under consideration require that each agent be aware of certain network-wise information, such as spectral properties of the underlying graph or total number of agents in the network [16, 17, 20–22]. Thus, the key technical challenge here is how to design a fully local upper confidence bound function for each agent, which does not require any global information. To achieve this, we aim to bound the variance proxy of each agent's local estimate of each arm's sample mean by a function of the agent's local sample counter, in contrast to a function of all $N$ agents' sample counters used in the existing literature. To this end, another salient challenge arises, due to information latency. Although each agent can directly or indirectly receive processed information from all other agents in a connected network, it takes extra time from the agents other than its neighbors. Thus, the information each agent receives does not reveal the "current" states of all other agents. Meanwhile, each agent may have different exploration trajectories of the arms. Information coupling and latency may further increase this exploration "imbalance" among the network, leading to poor learning performance of those agents with relatively insufficient exploration. This is a typical bottleneck of multi-armed bandit learning processes. To tackle this, we design a local decision making criterion which provably bounds the difference between each agent's local sample number and the maximal number of samples over the network. The criterion enables the explorations of each arm among all the agents approximately "on the same page" and thus gets around the "imbalance" bottleneck.

**Contributions**   We propose a fully decentralized multi-armed bandit algorithm for directed, strongly connected graphs, without requiring any global information. The algorithm is shown to guarantee that each agent achieves a better logarithmic asymptotic regret than the classic single-agent UCB1 algorithm. It appears that our work provides the first fully decentralized multi-armed bandit algorithm for directed graphs, with a provable regret guarantee for strongly connected graphs. Extensive simulations show that the algorithm also works for more general weakly connected graphs. For

the special case when the underlying graph is undirected, our algorithm can be modified to have a
further improved regret which reflects the effect of each agent's degree centrality in the graph. In this
case, the algorithm enables faster learning for any agent in the network contrasted with the UCB1
algorithm, as long as the agent has at least one neighbor.

## 2 Problem Formulation

As mentioned in the introduction, we are interested in a decentralized multi-armed bandit problem
formulated as follows. Consider a multi-agent network consisting of $N$ agents (or players). For
presentation purposes, we label the agents from 1 through $N$. It is worth emphasizing that the agents
are not aware of such a global labeling, but each agent can differentiate between its neighbors. The
neighbor relations among the $N$ agents are described by a directed graph $\mathbb{G} = (\mathcal{V}, \mathcal{E})$ with $N$ vertices,
where the vertex set $\mathcal{V} = [N] \triangleq \{1, 2, \ldots, N\}$ represents the $N$ agents and the set of directed edges
(or arcs) $\mathcal{E}$ depicts the neighbor relations where agent $j$ is a neighbor of agent $i$ whenever $(j, i)$
is a directed edge in $\mathbb{G}$. Each agent can receive information only from its neighbors (i.e., lies in
its neighbors' broadcast ranges). Thus, the directions of directed edges represent the directions of
information flow. For convenience, we assume each agent is a neighbor of itself, or equivalently, each
vertex of $\mathbb{G}$ has a self-arc. Clearly, a directed graph $\mathbb{G}$ may allow uni-directional communication
among the agents. In the case when $(i, j)$ is an edge in $\mathbb{G}$ as long as $(j, i)$ is an edge in the graph, $\mathbb{G}$
becomes an undirected graph which only allows bi-directional communication.

All $N$ agents face a common set of $M$ arms (or decisions) which is denoted by $[M] \triangleq \{1, 2, \ldots, M\}$.
At each discrete time $t \in \{0, 1, 2, \ldots, T\}$, each agent $i$ makes a decision on which arm to select from
the $M$ choices, and the selected arm is denoted by $a_i(t)$. If agent $i$ selects an arm $k$, it will receive a
random reward $X_{i,k}(t)$. For each $i \in [N]$ and $k \in [M]$, $\{X_{i,k}(t)\}_{t=1}^T$ is an unknown i.i.d. random
process. For each arm $k \in [M]$, all $X_{i,k}(t)$, $i \in [N]$, share the same expectation $\mu_k$. It is worth
emphasizing that this setting allows different agents to have different reward probability distributions
for each arm, so long as their means are the same. Without loss of generality, we assume that all
$X_{i,k}(t)$ have bounded support $[0, 1]$ and that $\mu_1 \geq \mu_2 \geq \cdots \geq \mu_M$, which implies that arm 1 has the
largest reward mean and thus is always an optimal choice.

The goal of the decentralized multi-armed bandit problem just described is to devise a decentralized
algorithm for each agent in the network which will enable agent $i$ to minimize its expected cumulative
regret, defined as

$$R_i(T) = T\mu_1 - \sum_{t=1}^T \mathbf{E}\left[X_{a_i(t)}\right],$$

at an order at least as good as $R_i(T) = o(T)$, i.e., $R_i(T)/T \to 0$ as $T \to \infty$, for all $i \in [N]$.

It is worth re-emphasizing that all the existing algorithms [16, 17, 20–22] for the above decentralized
MAB problem require each agent to make use of certain network-wise global information such as the
spectral properties of the neighbor graph or total number of agents in the network. In the next section,
we propose a fully decentralized multi-armed bandit algorithm which does not require any global
information.

## 3 Algorithm

We begin with some important variables to help present our algorithm.

**Local sample counter:** Let $n_{i,k}(t)$ be the number of times agent $i$ pulls arm $k$ by time $t$.

**Local sample mean:** Let $\mathbb{1}(\cdot)$ be the indicator function that returns 1 if the statement is true and 0
otherwise. Define

$$\bar{x}_{i,k}(t) = \frac{1}{n_{i,k}(t)} \sum_{\tau=0}^t \mathbb{1}(a_i(\tau) = k) X_{i,k}(\tau), \tag{1}$$

which represents the average reward that agent $i$ receives from arm $k$ until time $t$. This is analogous
to how a single agent estimates the reward mean in UCB1 [3].

**Two local estimates:** Each agent can have more sample information and a more accurate reward
mean estimate for each arm by exploiting information received from its neighbors, since all the agents

are simultaneously exploring the arms. To this end, each agent $i$ uses two variables, $m_{i,k}(t)$ and $z_{i,k}(t)$, to locally estimate two pieces of global information, the maximal number of samples of arm $k$ pulled among all the $N$ agents till time $t$, $\max_{j \in [N]} n_{j,k}(t)$, and the sample mean of arm $k$ among all the $N$ agents, respectively. At each time $t$, each agent $i$ updates its $z_{i,k}(t)$ and $m_{i,k}(t)$ as follows:

$$m_{i,k}(t+1) = \max\{n_{i,k}(t+1),\ m_{j,k}(t),\ j \in \mathcal{N}_i\}, \tag{2}$$

$$z_{i,k}(t+1) = \sum_{j \in \mathcal{N}_i} w_{ij} z_{j,k}(t) + \bar{x}_{i,k}(t+1) - \bar{x}_{i,k}(t), \tag{3}$$

where $\mathcal{N}_i$ denotes the set of neighbors of agent $i$ including itself, and $w_{ij}$, $j \in \mathcal{N}_i$, are "consensus" weights to be designed using local information only. It is worth emphasizing that both $m_{i,k}(t)$ and $z_{i,k}(t)$ are updated in a distributed manner as only information from agent $i$'s neighbors are needed.

The updates (2) and (3) are intended to reach an "approximate" agreement on the two estimates among the $N$ agents. The update (2) makes use of the idea of max-consensus [28]. The update (3) consists of two components, $\sum_{j \in \mathcal{N}_i} w_{ij} z_{j,k}(t)$, a linear consensus term, and $\bar{x}_{i,k}(t+1) - \bar{x}_{i,k}(t)$, which can be regarded as a local "gradient" term. Intuitively, $z_{i,k}(t)$ is a better estimate of the reward mean compared with the local sample mean $\bar{x}_{i,k}(t)$, as $z_{i,k}(t)$ exploits more sample information.

**Two local design objects:** Each agent $i$ needs to specify two objects in its local implementation. The first object is the consensus weights $w_{ij}$, $j \in \mathcal{N}_i$, which will be used in the update (3). Consensus algorithms have been studied for many years. We will appeal to two classic linear consensus processes: the flocking algorithm [29] and the Metropolis algorithm [30], tailored for consensus over directed graphs and average consensus over undirected graphs, respectively. The second object is the upper confidence bound function $C_{i,k}(t)$ which will be used to quantify agent $i$'s belief on its estimate of arm $k$'s reward mean. Upper confidence bound functions are critical in single-agent UCB algorithm design. As we will see, coordination among the agents allows us to design upper confidence bound functions "better" than that in the classic UCB1 algorithm [3]. Detailed expressions of the consensus weights and upper confidence bound functions will be specified in the theorems.

A detailed description of our decentralized UCB algorithm, named `Dec_UCB`, is presented as follows.

### 3.1 `Dec_UCB`: Decentralized UCB

**Initialization:** At time $t = 0$, each agent $i$ samples each arm $k$ exactly once, setting $m_{i,k}(0) = n_{i,k}(0) = 1$, $z_{i,k}(0) = \bar{x}_{i,k}(0) = X_{i,k}(0)$, and $C_{i,k}(0) = 0$.

**Iteration:** Between clock times $t$ and $t+1$, $t \in \{0, 1, \ldots, T\}$, each agent $i$ performs the steps enumerated below in the order indicated.

1. **Decision Making:** Each agent $i$ picks exactly one arm according to the following rule:
    (a) If there is no arm $k$ such that $n_{i,k}(t) \leq m_{i,k}(t) - M$, agent $i$ computes the index
    $$Q_{i,k}(t) = z_{i,k}(t) + C_{i,k}(t),$$
    and then pulls the arm $a_i(t+1)$ that maximizes $Q_{i,k}(t)$, with ties broken arbitrarily, and receives reward $X_{i,a_i(t+1)}(t+1)$.
    (b) If there exists at least one arm $k$ such that $n_{i,k}(t) \leq m_{i,k}(t) - M$, then agent $i$ randomly pulls one such arm.
2. **Transmission:** Agent $i$ broadcasts its $m_{i,k}(t)$ and $z_{i,k}(t)$; at the same time, agent $i$ receives $m_{j,k}(t)$ and $z_{j,k}(t)$ from each of its neighbors $j \in \mathcal{N}_i$.
3. **Updating:** Each agent $i$ updates the following variables for each arm $k$:

$$n_{i,k}(t+1) = \begin{cases} n_{i,k}(t) + 1 & \text{if } k = a_i(t+1), \\ n_{i,k}(t) & \text{if } k \neq a_i(t+1), \end{cases}$$

$$\bar{x}_{i,k}(t+1) = \frac{1}{n_{i,k}(t+1)} \sum_{\tau=0}^{t+1} \mathbb{1}(a_i(\tau) = k) X_{i,k}(\tau),$$

$$m_{i,k}(t+1) = \max\{n_{i,k}(t+1),\ m_{j,k}(t),\ j \in \mathcal{N}_i\},$$

$$z_{i,k}(t+1) = \sum_{j \in \mathcal{N}_i} w_{ij} z_{j,k}(t) + \bar{x}_{i,k}(t+1) - \bar{x}_{i,k}(t).$$

182 For a concise presentation of the algorithm, we refer to the pseudocode in Appendix A.

183 To better understand the algorithm just described, we provide the following remarks.

184 **Remark 1.** *In the special case when $N = 1$, i.e., the single-agent case, let agent $i$ be the unique*
185 *agent in the network. Clearly, there is no information transmission involved. Note that in this case,*
186 *$n_{i,k}(t)$ always equals $m_{i,k}(t)$, which implies that the inequality $n_{i,k}(t) > m_{i,k}(t) - M$ always holds.*
187 *Thus, at each time, the agent pulls an arm that maximizes $Q_{i,k}(t)$. Also, the update of $z_{i,k}(t)$ can be*
188 *simplified as $z_{i,k}(t+1) - z_{i,k}(t) = \bar{x}_{i,k}(t+1) - \bar{x}_{i,k}(t)$. Since $z_{i,k}(0) = \bar{x}_{i,k}(0)$, it follows that*
189 *the reward mean estimate $z_{i,k}(t)$ is always the same as the sample mean $\bar{x}_{i,k}(t)$. Therefore, `Dec_UCB`*
190 *is essentially the same as the classic single-agent UCB1 algorithm proposed in [3] when $N = 1$.* □

191 Since our decentralized UCB algorithm simplifies to the classic UCB1 [3] as explained in the
192 above remark, we will focus on our algorithm performance comparison with respect to UCB1, both
193 theoretically and experimentally. It is worth mentioning that [3] also proposes another single-agent
194 UCB algorithm, named UCB2.

195 **Remark 2.** *A key aspect of the algorithm design, which is different from classic single-agent UCB*
196 *algorithms and existing decentralized MAB algorithms [16, 17, 20, 21], is the inequality criterion*
197 *in the Decision Making rule (a), $n_{i,k}(t) \le m_{i,k}(t) - M$. The intuition behind this is to restrict*
198 *the difference between the local sample counter $n_{i,k}(t)$ and the local estimate $m_{i,k}(t)$. Since the*
199 *differences are uniformly bounded above by $M$, the inequality to some extent enables all the agents*
200 *to be "consistent" in exploring the arms, that is, no agent will be behind too much in exploring any*
201 *arm. The motivation in doing so is that a typical bottleneck of multi-armed bandits lies in insufficient*
202 *exploration of one or more arms. In our decentralized setting, if one agent does not sufficiently*
203 *explore an arm, it will affect the accuracy of the reward mean estimate of all other agents as the graph*
204 *is connected in some form. Keeping the explorations of each arm among all the agents approximately*
205 *"on the same page" gets around the bottleneck.* □

## 3.2 Results

207 To state our first result, we need the following concepts.

208 Let $z_k(t)$ and $\bar{x}_k(t)$ be the $N$-dimensional vectors whose $i$th entries equal $z_{i,k}(t)$ and $\bar{x}_{i,k}(t)$, respec-
209 tively. Then, the updates (3) for the $N$ agents can be combined as

$$z_k(t+1) = W z_k(t) + \bar{x}_k(t+1) - \bar{x}_k(t), \tag{4}$$

210 where $W$ is the $N \times N$ matrix whose $ij$th entry equals $w_{ij}$ if $j \in \mathcal{N}_i$ and zero otherwise. In the case
211 where each agent adopts the flocking algorithm [29], i.e., (5) in Theorem 1, $W$ is a stochastic matrix
212 whose diagonal entries are all positive. The flocking algorithm can be applied to both directed and
213 undirected graphs. In the case where each agent adopts the Metropolis algorithm [30], i.e., (8) in
214 Theorem 2, $W$ is a symmetric doubly stochastic matrix whose diagonal entries are all positive. The
215 Metropolis algorithm can only be applied to undirected graphs [30].

### 3.2.1 Strongly Connected Graphs

217 A directed graph is strongly connected if it has a directed path from any vertex to any other vertex.
218 For a strongly connected graph $\mathbb{G}$, the distance from vertex $i$ to another vertex $j$ is the length of the
219 shortest directed path from $i$ to $j$; the longest distance among all ordered pairs of distinct vertices $i$
220 and $j$ in $\mathbb{G}$ is called the diameter of $\mathbb{G}$.

221 Let $\Delta_k = \mu_1 - \mu_k$ for each $k \in [M]$, denoting the gap of reward means between arm 1 and arm $k$.

222 **Theorem 1.** *Suppose that $\mathbb{G}$ is strongly connected and all $N$ agents adhere to `Dec_UCB`. Then, with*

$$C_{i,k}(t) = \sqrt{\frac{4 \log t}{3 n_{i,k}(t)}} \quad and \quad w_{ij} = \frac{1}{|\mathcal{N}_i|}, \quad j \in \mathcal{N}_i, \tag{5}$$

223 *the regret of each agent $i \in [N]$ until time $T$ satisfies*

$$R_i(T) \le \sum_{k:\Delta_k > 0} \left( \max \left\{ \frac{16}{3\Delta_k^2} \log T, \ 2(M^2 + 2Md + d), \ L \right\} + \frac{2\pi^2}{3} + M^2 + (2M-1)d \right) \Delta_k,$$

224 *where $d$ is the diameter of $\mathbb{G}$, and $L$ is a constant defined in Remark 3.*

Here $|\mathcal{N}_i|$ denotes the cardinality of $\mathcal{N}_i$, or equivalently, the number of neighbors of agent $i$ including itself. Thus, $|\mathcal{N}_i|$ is always positive.

It is worth noting that the above regret bound intuitively decreases as the diameter of neighbor graph $\mathbb{G}$ decreases or the network connectivity increases (see Remark 3).

To better understand the above theorem, let $T$ be sufficiently large. Then, the regret bound in the theorem can be written as $\sum_{\Delta_k>0}(\frac{16}{3\Delta_K}+o(1))\log T$. Compared with the regret bound of the classic single-agent UCB1 given in [3], which is $\sum_{\Delta_k>0}(\frac{8}{\Delta_k}+o(1))\log T$, we have the following result.

**Corollary 1.** *If the neighbor graph is strongly connected, `Dec_UCB` guarantees each agent to achieve a better logarithmic asymptotic regret than the classic UCB1.*

**Remark 3.** *It can be seen that $W$ is an irreducible and aperiodic stochastic matrix (which holds for both Theorem 1 and Theorem 2). Then, it is well known that there exists a rank-one stochastic matrix $W_\infty$ for which $W^t$ converges to $W_\infty$ exponentially fast as $t \to \infty$ [31]. To be more precise, letting $\rho_2$ denote the second largest among the magnitudes of the $N$ eigenvalues of $W$, then $\rho_2 \in [0,1)$ and there exists a positive constant $c$ such that*

$$\left|\left[W^t\right]_{ij} - [W_\infty]_{ij}\right| \le c\rho_2^t \tag{6}$$

*for all $i,j \in [N]$, where $[\cdot]_{ij}$ denotes the $ij$th entry of a matrix. With the above $c$ and $\rho_2$, $L$ is defined as the smallest value such that when $t \ge L$, there holds*

$$72N\lceil c \rceil t \rho_2^{\frac{t}{12N\lceil c \rceil}-1} < 1, \tag{7}$$

*where $\lceil \cdot \rceil$ denotes the ceiling function.*

*Since $\rho_2$ is nonnegative, LHS of (7) is decreasing in terms of $t$ when $t$ is large enough and converges to 0 as $t \to \infty$, which implies that the inequality always holds after some finite time. Thus, $L$ must be nonnegative and uniquely exist by its definition. Also, LHS is a power function of $\rho_2$, thus it is increasing in terms of $\rho_2$, i.e., the smaller $\rho_2$ is, the smaller LHS would be, given a fixed $t$. In another aspect, the smaller $\rho_2$ is, the faster the LHS converges to 0 as $t \to \infty$, which implies that $L$ decreases as $\rho_2$ decreases. Since $\rho_2$ can be regarded as an index of connectivity of $\mathbb{G}$ with weight matrix $W$ in that the smaller $\rho_2$ is, the higher connectivity the network has, $L$ decreases as the network connectivity increases. In the special case when $\rho_2 = 0$, it is easy to verify that $L = 0$.* $\qquad\square$

**Proof Sketch of Theorem 1** The proof makes use of important properties of sub-Gaussian random variables (see Appendix B.1). Since any random variable with bounded support is sub-Gaussian, so is any $X_{i,k}(t)$. With this in mind, we write each $z_{i,k}(t)$ as a linear combination of a set of sub-Gaussian random variables $X_{j,k}(\tau), j \in [N], \tau \in \{0,1,\ldots,t\}$, which is also sub-Gaussian due to the additivity property of sub-Gaussian random variables. A particularly important property of a sub-Gaussian random variable $X$ with mean $\mu$ and variance proxy $\sigma^2$ is that $\mathbf{P}(|X-\mu| \ge a) \le 2e^{-\frac{a^2}{2\sigma^2}}$ holds for any non-negative $a$. To make use of this property, our next step is to estimate the variance proxy of $z_{i,k}(t)$, denoted by $\sigma_{i,k}^2(t)$. To this end, we first show that $\sigma_{i,k}^2(t)$ is bounded above by a function of all $N$ sample counters, $n_{j,k}(t), j \in [N]$, that is, $\sigma_{i,k}^2(t) \le f(n_{1,k}(t), n_{2,k}(t), \ldots, n_{N,k}(t))$. A critical technical challenge here is to bound the $f$ function with a local function, i.e., a function depending only on agent $i$'s local sample counter $n_{i,k}(t)$. To tackle this, we invoke the key algorithm step in `Dec_UCB`, which is the inequality criterion in the Decision Making rule (a), designed to ensure all the agents will be "consistent" in exploring each arm (see Remark 2). Using this "consistency", we are able to replace the $f$ function with a local function $g$ with which $\sigma_{i,k}^2(t) \le g(n_{i,k}(t))$. Substituting this function and the upper confidence bound $C_{i,k}(t)$ into the inequality of sub-Gaussian random variables mentioned above, we can show that the reward mean $\mu_k$ is within the range of the confidence interval of agent $i$'s local estimate $z_{i,k}(t)$ with high probability, i.e., $\mathbf{P}(|z_{i,k}(t) - \mu_k| \ge C_{i,k}(t)) = o(1/t)$, which is also the key idea of how we design $C_{i,k}(t)$. What remains is to apply the analysis of UCB1 [3] to further transform the upper confidence bound to the regret bound. Specifically, we are then able to bound $\mathbf{E}(n_{i,k}(t))$ by a uniform constant for all non-optimal arms. This and the fact that $R_i(T) = \sum_{\Delta_k>0} \mathbf{E}(n_{i,k}(T))\Delta_k$ yield the upper bound of agent $i$'s regret. $\qquad\square$

### 3.2.2 Undirected Graphs

Note that the regret bound in Theorem 1, derived for strongly connected graphs using the flocking algorithm weights, is independent of agent index $i$ and thus each agent's centrality. In the following

theorem, we will show that when the neighbor graph is undirected, we can have a better regret bound using the Metropolis algorithm weights, which shows how each agent's regret bound depends on the number of its neighbors, i.e., the degree centrality.

For an undirected, connected graph $\mathbb{G}$, the distance between two different vertices is the length of the shortest path connecting them, and the diameter of $\mathbb{G}$ is the longest distance among all pairs of distinct vertices in $\mathbb{G}$.

**Theorem 2.** *Suppose that $\mathbb{G}$ is undirected, connected, and all $N$ agents adhere to* `Dec_UCB`*. Then, with*

$$C_{i,k}(t) = \sqrt{\frac{3\log t}{|\mathcal{N}_i| n_{i,k}(t)}} \quad and \quad \begin{cases} w_{ij} & = & \frac{1}{\max\{|\mathcal{N}_i|, |\mathcal{N}_j|\}}, \quad j \in \mathcal{N}_i, \quad j \neq i, \\ w_{ii} & = & 1 - \sum_{j \in \mathcal{N}_i} \frac{1}{\max\{|\mathcal{N}_i|, |\mathcal{N}_j|\}}, \end{cases} \tag{8}$$

*the regret of each agent $i \in [N]$ until time $T$ satisfies*

$$R_i(T) \leq \sum_{k:\Delta_k > 0} \left( \max\left\{ \frac{12}{|\mathcal{N}_i|\Delta_k^2} \log T, \ 2(M^2 + 2Md + d), \ L \right\} + \frac{2\pi^2}{3} + M^2 + (2M - 1)d \right)\Delta_k,$$

*where $d$ is the diameter of $\mathbb{G}$, and $L$ is a constant defined in Remark 3.*

To better understand the above theorem, let $T$ be sufficiently large. Then, the regret bound in the theorem can be written as $\sum_{\Delta_k > 0} (\frac{12}{|\mathcal{N}_i|\Delta_k} + o(1)) \log T$. Comparing this bound with the asymptotic regret bound in Theorem 1, that is $\sum_{\Delta_k > 0} (\frac{16}{3\Delta_K} + o(1)) \log T$, it can be seen that the former is smaller than the latter if $|\mathcal{N}_i| \geq 3$, which leads to the following result.

**Corollary 2.** `Dec_UCB` *guarantees an agent to learn faster in an undirected, connected graph than when the graph is directed, strongly connected, as long as the agent has at least two neighbors excluding itself.*

Simulations for the case when an agent only has two neighbors excluding itself can be found in Section 4 and Appendix C.

Next we compare $\sum_{\Delta_k > 0} (\frac{12}{|\mathcal{N}_i|\Delta_k} + o(1)) \log T$ with the asymptotic regret bound of the classic UCB1 algorithm, that is $\sum_{\Delta_k > 0} (\frac{8}{\Delta_k} + o(1)) \log T$. The former is smaller than the latter if $|\mathcal{N}_i| > 1$. Since each agent always has itself as a neighbor by assumption, and each agent must have at least one neighbor excluding itself in a connected graphs, $|\mathcal{N}_i| > 1$ always holds in a connected graph. We are led to the following result.

**Corollary 3.** *If the neighbor graph with $N > 1$ agents is undirected and connected,* `Dec_UCB` *guarantees each agent to achieve a better logarithmic asymptotic regret than the classic UCB1.*

Note that any undirected graph can always be divided into one or more connected components. Thus, each connected component can be analyzed separately and independently. Corollary 3 has the following immediate consequence.

**Corollary 4.** *If the neighbor graph is undirected,* `Dec_UCB` *guarantees an agent to achieve a better logarithmic asymptotic regret than the classic UCB1, as long as the agent has at least one neighbor excluding itself.*

### 3.2.3 Weakly Connected Graphs

Corollary 4 shows that when the neighbor graph is undirected, `Dec_UCB` is still functional with provable performance even if the graph is disconnected. However, the case of directed graphs is much more complicated. A disconnected directed graph can also be divided into more than one "connected" component, yet each component is "weakly connected", and not necessarily strongly connected. A directed graph is weakly connected if replacing all of its directed edges with undirected ones results in a connected graph. A strongly connected graph is weakly connected, but not vice versa. Thus, our results in Section 3.2.1 cannot be applied to weakly connected graphs, whose complete analysis has so far eluded us. Notwithstanding this, it is worth noting that simulations in Section 4 suggest that `Dec_UCB` works well for weakly connected graphs, and thus also for any directed graphs, as long as each agent has at least one neighbor excluding itself.

## 4 Simulations

This section presents various simulations created with the aid of Python packages [32–35] which were used to experimentally verify the validity and performance of our proposed `Dec_UCB` algorithm. We focus on the heterogeneous reward distribution case here. Additional simulations and observations, including the homogeneous reward distribution case, are presented in Appendix C.

**Small-size Graphs**  Simulations were run on three types of graphs, namely strongly connected, undirected connected, and weakly connected graphs[1], allowing for the reward distribution to vary between agents for a given arm. A given agent and arm pair can draw rewards from an arm-specific Beta distribution with mean $\mu_k$ and standard deviation 0.05, an arm-specific Bernoulli distribution with mean $\mu_k$, or an arm-specific truncated normal distribution within $[0, 1]$ with mean $\mu_k$ and standard deviation 0.05. The distribution used is randomly assigned to each agent/arm pair upon initialization. The reward means $\mu_k$ are the same for all agents on a given arm, with each $\mu_k$ randomly chosen from a uniform distribution on $[0.05, 0.95]$. Rewards are bounded by definition from the used distributions to be within $[0, 1]$. Each experiment is run for $T = 1000$ time steps with results for each graph obtained by averaging over 100 experiments. The results obtained from running `Dec_UCB` alongside UCB1 on small 6-agent graphs are illustrated in Figures 1, 2, and 3 for the three graph types, respectively. These results empirically back the claims of Theorem 1, Theorem 2, and Corollaries 1–3 in the presence of heterogeneous arm reward distributions.

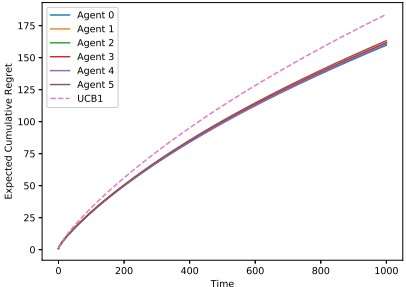
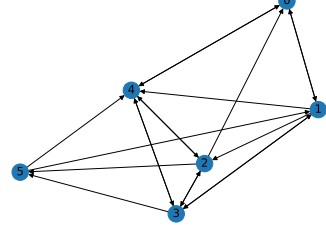

Figure 1: A plot of the regret of the strongly connected graph, averaged over 100 experiments. Reward distributions vary between agents for a given arm.

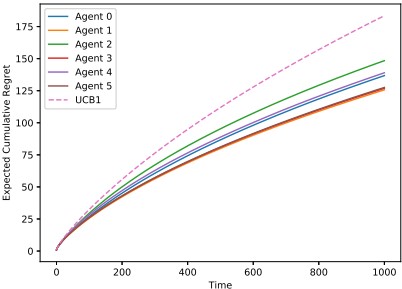
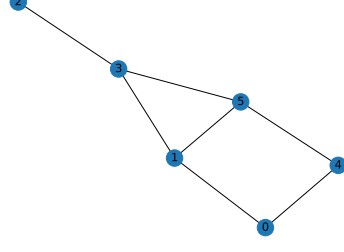

Figure 2: A plot of the regret of the undirected and connected graph, averaged over 100 experiments. Reward distributions vary between agents for a given arm.

**Large-scale Graphs**  Larger scale simulations were run for the three graph types with heterogeneous reward distributions, averaging results from 100 different randomly generated Erdős–Rényi 50-agent graphs for each graph type. Additionally, 10 arms with rewards following a randomly chosen Beta, Bernoulli, or truncated normal distribution were used, with means $\mu_k$ randomly chosen from a uniform distribution bounded within $[0.05, 0.95]$. A standard deviation of 0.05 was used for the Beta and truncated normal distributions. Rewards were bounded by the used distributions to be within $[0, 1]$. The algorithms were run for $T = 1000$ iterations for each different random graph, testing the

---

[1]We focus on weakly connected graphs in which each agent has at least one neighbor excluding itself; otherwise the agent cannot receive any external information and thus essentially lies in the single-agent case.

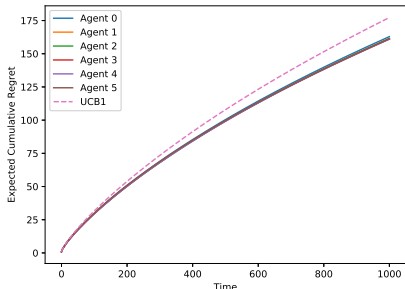 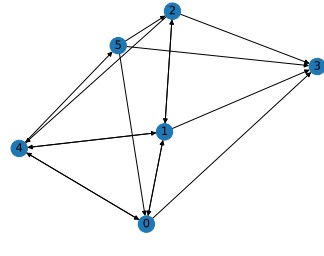

Figure 3: A plot of the regret of the weakly connected graph, averaged over 100 experiments. Reward distributions vary between agents for a given arm.

worst performing agent of `Dec_UCB` against the best performing results from the UCB1 algorithm. Results are shown in Figure 4. In all cases, `Dec_UCB` achieves better performance than UCB1.

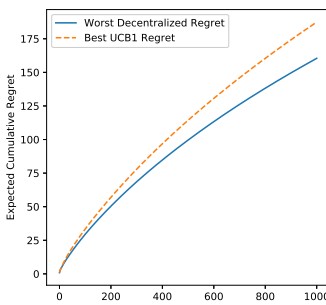 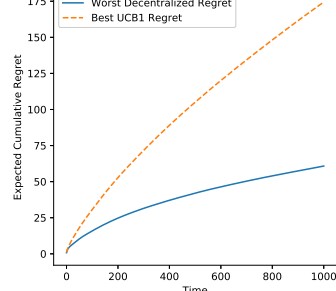 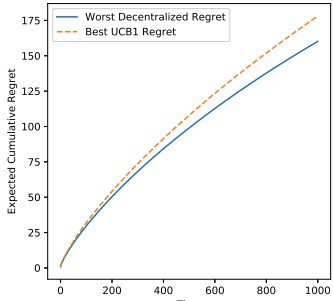

(a) Results for the large strongly connected generated graphs.

(b) Results for the large undirected connected generated graphs.

(c) Results for the large weakly connected generated graphs.

Figure 4: Plots of the expected cumulative regret for both the worst performing agent of `Dec_UCB` and best performance of UCB1. Results averaged over 100 different randomly generated Erdős–Rényi weakly connected graphs of 50 agents each. The reward distribution for a given arm was randomly chosen as a Beta, Bernoulli, or truncated normal distribution.

**Observations** There are several key observations to take from these simulations. The first of these is that `Dec_UCB` appears to perform better on the undirected graphs than it does on the strongly connected graphs. This validates the theoretical results presented in Theorem 2 and Corollary 2. Additionally, the performance of each agent in the strongly connected graphs appears to be independent of its number of neighbors, indicating that performance is reliant only on the diameter of the graph as demonstrated in Section 3.2.1. In contrast, as shown in Section 3.2.2, for undirected graphs, each agent's regret also depends on its number of neighbors. In total, performance of `Dec_UCB` appears to be unaffected by the choice of using homogeneous arm rewards or heterogeneous arm rewards; expected cumulative regrets appear to be nearly equivalent for all graph types in either case.

# 5   Conclusion

In this paper, we have studied a decentralized multi-armed bandit problem over directed graphs and proposed a fully decentralized UCB algorithm, which provably achieves a better logarithmic asymptotic regret than the classic UCB1 algorithm provided the neighbor graph is strongly connected. We have further improved the algorithm's performance for undirected graphs. Simulations have been provided to validate our theoretical results and test the performance on more general weakly connected graphs. Future directions are to study the limitations of the paper, including analysis for weakly connected graphs and time-varying graphs, experiments with real data sets, and development of a decentralized counterpart for the classic UCB2 algorithm [3].

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
