# OpenReview forum: "Decentralized Multi-Armed Bandit Can Outperform Classic Upper Confidence Bound"
_NeurIPS.cc/2021/Conference — NeurIPS 2021 Submitted_

### Official Review · Reviewer_w9xA · 2021-07-13

**Rating:** 6
**Confidence:** 4

**Summary:**

The paper considers a multi-agent cooperative bandit system, where the agents are connected on a graph and can share messages. The arm-pulls of each agent does not interfere with that of the others, but agents can speed up learning by communicating. The paper proposes a parameter-free (no information on the global system) algorithm that each individual agent executes locally. if the graph is strongly connected, the per-agent regret is shown to be strictly better than that of vanilla UCB. If the graph is undirected, the regret of an agent is inversely proportional to the number of neighbors it can share information with. The key algorithmic innovation is to ensure that the number of times an arm is pulled by all agents are within M (a constant which is a hyper-parameter) of each other. This enables agents to arrive at consensus and get low regret

**Limitations And Societal Impact:**

See above.

**Main Review:**

The paper is very well written and was a pleasure to read and follow along. Multi-agent bandit algorithms that are truly parameter-free are an important line of work and the present paper makes progress on that front.

My main question is whether in some case (a class of undirected graphs), can the algorithm achieve a regret reduction by a factor of N (the total number of agents), as opposed to N_i, the number of neighbors. Indeed for the complete graph they are the same, but can they be extended to for example expanders or other fast mixing topologies?

Related to above is a Missing reference :  The article of "The gossiping insert-eliminate algorithm for multi-agent bandits by Chawla et.al. in AISTATS 2021" is relevant to your study. They consider an algorithm that requires knowledge of number of agents, but obtains regret scaling as 1/N times that of the vanilla UCB, "for any undirected graph". Thus the regret guarantee is stronger compared to the present paper, but relies on global information.

More generally, I recommend a survey in the related work along the following lines -" stronger results have been obtained under global information while the present work obtains weaker results without any assumption of global information."

**Time Spent Reviewing:**

1.5

---

> ### Author Response · Authors · 2021-08-10
> **Response to Reviewer w9xA**
>
> First thanks for the very positive comments. As for your question, The answer is positive. Our regret for undirected graphs only achieves a reduction by a factor of $|\scr N_i|$ because we used $|\scr N_i|$ in the design of the upper confidence bound $C_{i,k}(t)$; we could not use $N$ as the network size is a global information. If we relax the ‘‘fully decentralized'' requirement and allow each agent to be aware of the network size $N$, we can replace $|\scr N_i|$ by $N$ in $C_{i,k}(t)$ in the algorithm. Then the variance proxy in Lemma 8 (in Appendix B.4) will be divided by $N$ instead of $|\scr N_i|$ (simply by deleting the last inequality in the proof), then we can replace $|\scr N_i|$ by $N$ in the design of $C_{i,k}(t)$. With this, our analysis will yield a regret at the order of $O((\log T)/N)$ for any connected graph. Thanks for the suggestion and letting us be aware of the missing reference. We will add the reference and a survey as you suggested in the final version.

---

> > ### Comment · Reviewer_w9xA · 2021-08-28
> > **Rigorous Comparison with existing work can help**
> >
> > I would also encourage the authors to consider comparisons through simulations with non-trivial baselines to further clarify the message of the paper. The simulations also allows you to control variety of factors in the environment (such as graph sparsity, connectivity etc. ) and compare your methods with other stronger baselines.

---

### Official Review · Reviewer_oU5f · 2021-07-16

**Rating:** 5
**Confidence:** 3

**Summary:**

The authors propose a novel algorithm in order to extend the UCB1 algorithm to a distributed setting where agents are represented by the nodes of a digraph and these agents are allowed to exchange information with all of their neighbors at each step. The authors prove two performance guarantees, one for a general connected digraph, another one when the graph is non directed.

**Main Review:**

1) The communication model is never really clearly stated. What are the agents allowed to exchange ? Are they forced to communicate with each neighbor at each time step ? How many bits are allowed to be exchanged per round ? This is fundamental in order to assess the applicability of the results: for instance it seems that the algorithm requires to exchange real numbers (as opposed to integers) between agents, which is impossible using a communication network unless some quantization is performed. Also communicating with all neighbors all the time might generate a lot of unnecessary traffic etc.

2) The authors do not provide information theoretic lower bounds on the regret of any algorithm in their setting, nor do they try to compare their performance guarantees to other lower bounds. Therefore it is hard to tell whether or not the performance guarantees of Theorem 1 are interesting.

3) Theorem 1 shows only a constant improvement with respect to UCB1, so that the regret of each node scales as $R_i(T) = O(M\log(T)/\Delta)$ even though nodes are allowed to exchange a lot of information. One would expect the regret of each node to scale as $R_i(T) = O((M/N) \log(T)/\Delta )$ simply based on the argument that, if all agents explore an arm $(1/N) \log(T)/\Delta^2$ times and then broadcast their respective estimates to other nodes (one can always broadcast information to all nodes in at most $N$ communication rounds, providing that the graph is connected), then this arm can be safely discarded.

4) In Theorem 2, the choice of weights implies that all nodes know the number of their neighbors, as well as the number of their neighbors neighbors. This contradicts the assumptions in the model (unless one first designs an initial step in the algorithm in order to exchange this information as well).

5) Why not use KL-UCB instead of UCB in the algorithm's design ? Especially since they require the same information. Also, the numerical performance of the proposed algorithm should be compared to that of KL-UCB and Thompson sampling, which are the state-of-the-art algorithms for the centralized case.

6) In Theorem 1, the regret bound features a term proportional to $M^2 \sum_{k} \Delta_k$ which can scale like $O(M^3)$. This suggests that the algorithm can sometimes perform much worse than UCB1 since its regret scales like $O(M \log(T)/\Delta )$. Why is that so ? Is this an artifact of the analysis or does this phenomenon really appear in numerical experiments ?

7) In Theorem 1, a number $L$ appears, and Remark 3 bounds this number as a function of the eigenvalue gap of the graph. However, how does $L$ scale with $N$ ? Does there exists networks with arbitrarily large L value (if so this is a problem) ? What is the "worse case" network ?

Minor Comments: the authors claim that their approach is "fully distributed" which is misleading since a "full distributed" setting would be the case where agents are not allowed to exchange any information. Here they do exchange information with their neighbors.


**Time Spent Reviewing:**

2

---

> ### Author Response · Authors · 2021-08-10
> **Response to Reviewer oU5f**
>
> 1. We noticed that there exist different ways to introduce the model/problem in the literature. For example, [17,22] introduced the problem/model in the same way as ours (actually we followed this way), and [12] describes the problem using the ‘‘communication protocol''. The latter way  specifies information to be exchanged in the ‘‘communication protocol'', whereas we specify this in the algorithm description. We will clarify this point in the final version to avoid possible confusion, or happy to describe the problem formulation using another way which the reviewer feels better.
>
>
> In our model, each agent sends communicates with each of its neighbors (transmitting 2$M$ scalars) at each time step. Such a scheme is sometimes called ‘‘broadcast'' in the literature, and widely used in decentralized MAB (e.g. [12, 17]). Communication-efficient schemes (like ‘‘gossiping'' [7, 23]) are definitely interesting future directions.
> We agree that quantized communication is a fundamental issue and worth studying as another future direction. The main purpose of this paper is to design the first fully decentralized MAB algorithm, assuming broadcast and perfect communication. We would like to point out that many existing papers assume broadcast and perfect communication, including the published NIPS ones e.g. [12, 17], and that quantized communication and communication efficiency are often treated in separate papers for other distributed algorithms (e.g. distributed optimization and reinforcement learning).
>
> 2. We can provide the following lower bounds for our decentralized algorithms.
>
> For any strongly connected graph, our algorithm with (5) has a lower bound $\Omega((\log T)/N)$ on each agent's regret. The proof sketch is as follows:
>
> Among all strongly connected graphs with $N$ agents, a complete graph leads to the best network regret (i.e., the sum of all $N$ agents' regrets) because each agent can receive information from all the agents. This is essentially equivalent to a single-agent case in which the agent can pull N times at each time. It is well known that the (asymptotic) lower bound of the classic single-agent UCB1 (the agent pulls exactly once every time) is $\Omega(\log T)$. Then, with pulling $N$ times at each time, the regret lower bound becomes $\Omega(\log NT)$. Since this regret equals to the network regret of the complete graph case and all $N$ agents are homogeneous, each agent's regret has a lower bound $(1/N)\Omega(\log NT)$, which asymptotically equals $\Omega((\log T)/N)$.
>
> Similarly, the same lower bound applies to any undirected, connected graph with (8). This lower bound can be tight if we allow each agent to be aware of the network size $N$. In this case, we can replace $|\scr N_i|$ by $N$ in $C_{i,k}(t)$ in the algorithm. Then the variance proxy in Lemma 8 (in Appendix B.4) will be divided by $N$ instead of $|\scr N_i|$ (simply by deleting the last inequality in the proof), then we can replace $|\scr N_i|$ by N in the design of $C_{i,k}(t)$. With this, our analysis will yield a regret at the order of $O((\log T)/N)$ for any connected graph.
>
>
> We will add the above lower bound result in the final version.
>
> Considering the above lower bound, and as commented by Reviewer LxWs in his/her point 3, the improvement of decentralized UCB1 over centralized UCB1 can necessarily come only on the constant in front of the $\log(T)$ term. Our Theorem 1 achieves this improvement, and we thus believe it is an interesting result (see our response to point 3 by Reviewer LxWs for more details).
>
> 3. In our algorithm, each agent only needs to transmit 2$M$ scalars to its neighbors where $M$ is the number of arms. If we allow each agent's estimate to be ‘‘relayed'' with $N$ (or graph diameter) rounds, each agent needs a unique ID (a global ordering), and the set of information to be transmitted will be at the order of $MN$, which is not scalable and thus adding a very big burden in communication. In addition, the network size $N$ or graph diameter is a global information, which is not known to each agent; thus each agent will not be able to figure out whether all agents' information has been collected or not.
>
>
> From what we understand, your argument for achieving $O((M/N)\log(T))$ regret is based on the assumption that each agent is aware of $N$. As explained in our response to your point 2, with the same assumption, we can also achieve $O((M/N)\log(T))$ regret by slightly modifying our algorithm for any undirected, connected graphs, which is consistent with what you expected.
>
>
> We cannot understand why ‘‘this arm can be safely discarded''. Since the reward is a random variable, agents cannot discard an arm based on a finite number of samples; indeed, this is why we need to ensure sufficient exploration on each arm. It is possible that we missed your point here, and we are happy to respond after you clarify this point.
>
> 4. Your understanding of the weights design in Theorem 2 is correct. Each agent needs to send the number of its neighbors, to each of its neighbors (only) at the initial step. Note that this number is still a local information. If you can tell us to which assumption/setting this design conflicts, we will definitely make changes accordingly to avoid possible confusion.
>
> 5. The main purpose of this paper is to show that a (actually the first) fully decentralized multi-armed bandit algorithm can be crafted and outperform its classic single-agent counterpart. We achieved this by focusing on UCB1 because of two reasons: first, many existing UCB algorithms are based on UCB1; second, all existing UCB-based decentralized algorithms require global information. With these in mind, we respectfully disagree with the reviewer in that the comparison should be made between a decentralized algorithm and its centralized counterpart; otherwise the comparison will be unfair.
>
>
> We agree that decentralized KL-UCB and Thompson sampling are very interesting future directions and worth studying, and our decentralized UCB1 shows that they are promising.
>
> 6. The reviewer tried to compare a constant term and a $\log T$ term. Our goal, as in our theorems, is to show that the asymptotic regret (i.e. the $\log T$ term) of our decentralized algorithm is better (in constant) than that of the classic UCB1; see point 3 by Reviewer LxWs for validation. It is true that if we compare in the case when $T$ is small and then the constant terms matter, our theoretical regret upper bound can be larger than UCB1 with large $M$; however, our simulations show that our decentralized algorithm always outperforms UCB1 in real regret.
>
> 7. There is no explicit relation between $L$ and $N$. Note that $L$ is also dependent on graph connectivity $\rho_2$.
> We did extensive simulations during the rebuttal period, which shows that $L$ increases at most linear with $N$. That is to say, as long as the network size is finite, so is $L$, which can also be seen from (7). Our results and analyses are all based on the fact that the network size is finite (analysis of infinite network is completely a different story), so $L$ cannot be arbitrarily large. Even though $L$ may yield a large constant term in theoretical regret upper bound, our simulations show that our decentralized algorithm always outperforms UCB1 in real regret for any graphs.
>
> 8. As for the minor comments, in the existing literature, ‘‘distributed'' and ‘‘decentralized'' sometimes refer to different settings, sometimes are used interchangeably. We call our algorithm ‘‘fully decentralized'' in the sense that each agent only utilizes ‘‘local'' information -- its own information and information from its (one-hop) neighbors, in contrast to existing algorithms which require at least one piece of ‘‘global'' information (e.g. the network size or unique ID).

---

### Official Review · Reviewer_HFTD · 2021-07-25

**Rating:** 5
**Confidence:** 4

**Summary:**

A multi-agent pure-distributed (no shared data) algorithm for the stochastic Multi Armed Bandit setting is presented and analyzed.
Regret bounds for different agents connectivity schemes are provided, incorporating the related quantities, namely the diameter for strongly connected graphs and the number of each agent neighbors for undirected graphs. Experiments are conducted to illustrate the superiority of the algorithms and the related bounds over the single-agent UCB1 algorithm performance and the qualitative difference of the performance of the agents in the different connectivity settings.

**Limitations And Societal Impact:**

yes

**Main Review:**

The main contribution and novelty of the paper is the pure distributed algorithm and the regret bounds for the different connectivity settings. The (synchronous) algorithm presented is simple and easy to implement, the results are clearly stated, and the experiments nicely illustrate the claims.  The asymptotic performance advantage over the classical UCB1 (or other algorithms in which agents do not share information) is not surprising, nevertheless. Furthermore, the proof sketch of Theorem 1 in the main body of the paper does not provide any insight regarding the incorporation of the characteristics of the connectivity architecture (namely, the diameter of the strongly connected graph) or the constant $L$ of remark 3 into the regret bound (the main contribution of the paper).
Theory-wise,  beyond the improvement in the constant factor $\log T$ (from 8 to $\frac{16}{3}$), also (and probably more) interesting would be a lower bound on the achievable improvement (with a multiplicity of $N$ agents sampling the bandits) and the relation to the better bound of UCB2 (which is indeed mentioned in line 194 but without commenting on the performance or the applicability of the analysis).

typo: should be $X_{i, a_{i}(t)}$ in the formula after line 129.

omission? rewards after "arm k" in line 147.

The main algorithm is clearly presented, probably with some redundancy as the updates appear several times (e.g., after line 148, in  the algorithm description after line 181 and in the pseudocode in appendix A).

**Time Spent Reviewing:**

3

---

> ### Author Response · Authors · 2021-08-10
> **Response to Reviewer HFTD**
>
> 1. First thanks for the positive comments.
>
> 2. As pointed out, actually two things are reasonable to expect for the current homogeneous reward setting. First, the algorithm is reasonably expected to be fully decentralized because each agent can independently learn the optimal arm without any neighbor's information. Second, each agent's regret is reasonably expected to improve (in constant) since coordination and communication between neighboring agents allow each agent to collect more information at each time instant. However, all the existing algorithms require that each agent be aware of certain network-wise global information; specifically, they make use of either spectral properties of the underlying communication graph, or the total number of agents in the network, or global ordering (i.e., each agent has a unique identification number). This leads to a counterintuitive observation in the existing literature -- more information collected at each agent unexpectedly makes the distributed bandit algorithm design more restrictive. This is precisely what we want to resolve in this paper. Our main contributions are to propose a fully decentralized UCB algorithm and prove its performance is better than the classic single-agent UCB (in the way as the reviewer expects).
>
> 3. As for the writing for the Proof Sketch of Theorem 1, we focused on the key steps which reflect the technical challenges in the proofs. How these steps involve the graph diameter and the constant $L$ is implicitly reflected in the detailed proofs provided in the appendix.
> Specifically, the graph diameter is used in Appendix B.2 to establish ‘‘exploration consistency'', and $L$ is needed to estimate the variance proxy of the reward mean estimate in Appendix B.3.
> We will explicitly clarify the connection between the key steps and those important parameters (i.e., the graph diameter and the constant $L$) in the final version.
>
> 4. We can provide the following lower bounds for our decentralized algorithms.
>
> For any strongly connected graph, our algorithm with (5) has a lower bound $\Omega((\log T)/N)$ on each agent's regret. The proof sketch is as follows:
>
> Among all strongly connected graphs with $N$ agents, a complete graph leads to the best network regret (i.e., the sum of all $N$ agents' regrets) because each agent can receive information from all the agents. This is essentially equivalent to a single-agent case in which the agent can pull $N$ times at each time. It is well known that the (asymptotic) lower bound of the classic single-agent UCB1 (the agent pulls exactly once every time) is $\Omega(\log T)$. Then, with pulling $N$ times at each time, the regret lower bound becomes $\Omega(\log NT)$. Since this regret equals to the network regret of the complete graph case and all $N$ agents are homogeneous, each agent's regret has a lower bound $(1/N)\Omega(\log NT)$, which asymptotically equals $\Omega((\log T)/N)$.
>
>
> Similarly, the same lower bound applies to any undirected, connected graph with (8).
>
> We will add the above lower bound result in the final version.
>
> 5. The main purpose of this paper is to show that a fully decentralized multi-armed bandit algorithm (the first of its kind) can be crafted and outperform its classic single-agent counterpart. We achieved this by focusing on UCB1, as many existing UCB algorithms are based on UCB1.
> We agree that extending the results here to UCB2 is an interesting and important future direction, as mentioned in the Conclusion section, and believe that our work on UCB1 paves a way toward this extension.
>
> Thanks very much for pointing out the typo and omission. We will correct them in the final version. We will also polish the algorithm description to avoid the redundancy.

---

### Official Review · Reviewer_LxWs · 2021-07-26

**Rating:** 5
**Confidence:** 3

**Summary:**

This paper studies the classic multi-armed bandit problem, but in the presence of many cooperating agents distributed over a graph.

**Limitations And Societal Impact:**

Yes

**Main Review:**

This paper studies the classic multi-armed bandit problem, but in the presence of many agents distributed over a graph. Each agent is a node in a graph, and can receive messages from the neighbors of that node. Every agent is faced with making a decision in every time step, for t = 1...T. Each agent wants to minimize regret, and the goal is to have even the agent with the largest regret suffer a smaller regret than the standard single-agent UCB algorithm. This can be expected because after all an agent comes to possess more information now with the gossipping that goes on because of the network connections.

This area of cooperating agents on a network playing MAB has seen decent activity recently, with papers developing methods to achieve exactly this objective of smaller regret than the single-agent case. This paper's goal is to ensure that each agent knows nothing more than their neighbors about the graph --- not the total number of nodes or spectral properties of the graph etc. --- and achieve the same. This would make the cooperation model truly decentralized. The paper achieves this with a variant of UCB called Decentralized_UCB (Dec_UCB).

Evaluation:

1) Overall, at least theoretically, it makes an interesting question to ask what happens if agents cooperate in a network and exchange feedback information. So that's a plus.
2) But the model itself is not being introduced in this paper. It has been studied in many papers recently, and the paper's main contribution is to make it fully decentralized.
3) The improvement over UCB can necessarily come only on the constant in front of the log(T) term, and not asymptotically on the log(T) term itself. Also, regarding the improvement in constant, although there is *definitely* work to be done to establish this, overall, given the nature of the setting, it is reasonable to expect that an improvement in constants is going to happen when agents gossip over the network. Therefore, the results are not super surprising.
4) One interesting part is how the improved constant for the directed graphs case is independent of the number of neighbors, where as it is dependent on the number of neighbors for the undirected graphs case. The authors say that this is also borne out in simulations, although I don't exactly see the simulation plots that vary the number of neighbors.

**Time Spent Reviewing:**

5

---

> ### Author Response · Authors · 2021-08-10
> **Response to Reviewer LxWs**
>
> 1. Thanks for the positive comment.
>
> 2. We provided the multi-agent model and our goal (i.e., a fully decentralized algorithm) in the Problem Formulation section.
> Meanwhile, we noticed that there exist different ways to introduce the model/problem in the literature. For example, [17,22] introduced the problem/model in the same way as ours
> (actually we followed this way), and [12] describes the problem using the ‘‘communication protocol''. The latter way  specifies information to be exchanged in the ‘‘communication protocol'', whereas we specify this in the algorithm description. We will clarify this point in the final version to avoid possible confusion, and are happy to describe the problem formulation in a way more to the reviewer's satisfaction.
>
> 3. We totally agree with this and this is exactly the motivation of our paper. Actually two things are reasonable to expect for the current homogeneous reward setting. First, the algorithm is reasonably expected to be fully decentralized because each agent can independently learn the optimal arm without any neighbor's information. Second, each agent's regret is reasonably expected to improve (in constant) since coordination and communication between neighboring agents allow each agent to collect more information at each time instant. However, all the existing algorithms require that each agent be aware of certain network-wise global information; specifically, they make use of either spectral properties of the underlying communication graph, or the total number of agents in the network, or global ordering (i.e., each agent has a unique identification number). This leads to a counterintuitive observation in the existing literature -- more information collected at each agent unexpectedly makes the distributed bandit algorithm design more restrictive. This is precisely what we want to resolve in this paper. Our main contributions are to propose a fully decentralized UCB algorithm and prove its performance is better than the classic single-agent UCB (in the way as the reviewer expects).
>
> 4. For (strongly connected) directed graphs, our analysis shows that each agent's regret bound decreases as the diameter of neighbor graph decreases or the network connectivity increases (see Theorem 1 and Remark 3), but the improved constant (in front of the $\log T$ term) is independent of the number of neighbors. Figures 1 and 12 illustrate this.
>
> Moreover, Figure 3 shows how, similar to strongly connected graphs, agent regret in a weakly connected graph also seems to be independent of number of neighbors, as evidenced by how the regret curves for each agent belong to the same tight ‘‘band'' and are often indistinguishable from each other.
>
> For undirected graphs, we validated the regret's dependency on the number of neighbors (Theorem 2) in Figure 2. Note that for the given graph, Agent 2 has the smallest number of neighbors (only one excluding itself), Agents 0 and 4 have two neighbors, and Agents 1, 3, 5 have the largest number of neighbors (three neighbors). It can be seen in the plot that the regret of Agent 2 is the largest, the regrets of Agents 1, 3, 5 are very close and the smallest, and the regrets of Agents 0 and 4 are close and in the middle. All their regrets are smaller than the classic single-agent UCB. The same is also illustrated in Figures 11 and 13 in the appendix.
>
> We performed more simulations for large-scale graphs (more than 20 agents) during the rebuttal period, which are consistent with what we claimed, and will add them in the final version/appendix.

---

### Author Response · Authors · 2021-08-29
**Additional Response**

We thank Reviewer w9xA for his/her additional suggestion (to provide more simulation comparisons).

Please let us take this opportunity to clarify our motivation and contributions, which are related to the suggestion.
The contributions of our paper are two-fold. First, motivated by the fact that all existing decentralized algorithms require each agent to be aware of at least one piece of global information (e.g., the total number of agents, network connectivity, a unique ID number, an upper bound of all arms' variance proxies), we proposed the first (to our knowledge) fully decentralized multi-armed bandit algorithm based on UCB1. We chose UCB1 because it is one of the most classic bandit algorithms and quite a few existing (non-fully) decentralized algorithms are also based on it (e.g. [16, 20--23]).
Second, we proved that our proposed decentralized UCB1 outperforms UCB1 in terms of the asymptotic regret. The two contributions together deliver the message that fully decentralized design of bandits is not only possible, but can also have an advantage over centralized counterparts. Of course we only showed this for UCB1, but we believe it is a very promising starting point for many other classic and state-of-the-art bandit algorithms, e.g. UCB2 and KL-UCB. We treat these extensions as our future directions, as it is probably impossible to prove different decentralized bandit algorithms in one paper. An analogy is the area of distributed optimization in which each optimization method/algorithm has a series of papers focusing on its decentralized design. We will clarify these points in our final version, and are willing to modify the title to avoid possible over-claim.

Next we discuss the suggested comparisons. When we wrote the paper and rebuttals, we thought it unfair to compare decentralized UCB1 with other bandit algorithms. While it maybe debatable, after careful thoughts as both Reviewers w9xA and oU5f raised this issue, we actually can directly do the comparisons using our theoretical results. Specifically, for any undirected graph, our Theorem 2 shows that each agent's asymptotic regret is of the order $O(\frac{\log T}{|\mathcal{N}_i|})$ where $|\mathcal{N}_i|$ denotes the number of its neighbors. Since both UCB2 and KL-UCB have the regret bound $O(\log T)$, it is clear that our decentralized UCB1 can outperform UCB2 and KL-UCB as long as an agent has sufficient neighbors.
Our additional simulations also validate this.
Both our current simulations (in the paper and appendix) and additional simulations also suggest that graph connectivity has at least a mild effect on regrets.
Such comparisons for strongly connected graphs, or more general directed graphs, seem much more complicated and thus needs more careful analyses, but simulations are easy to perform. We will add the above discussion and simulations in the final version/appendix.

We did not do any comparisons with existing decentralized bandit algorithms because they all make use of global information; but we are willing to do so if any reviewer thinks it is worth doing. Meanwhile, any suggested (stronger) baselines by any reviewer are very welcomed and appreciated; we are happy to add them in the final version/appendix as additional simulations and discussions, which could be another contribution as the first set of comprehensive simulation comparisons (to the best of our knowledge such simulation comparisons in the existing literature are somewhat limited).

---

### Decision · Program_Chairs · 2021-09-27

**Decision:**

Reject

**Comment:**

The reviewers have all felt that while this is a decent paper, its contribution is somewhat uninspiring: the model it studies is nice but not new, and the improvement it provides over UCB is marginal and not very surprising.  The paper might benefit from a revision that will include stronger motivation and a more rigorous comparison with existing work---I suggest the authors to follow the suggestions of Reviewer w9xA to this end.